# Phase-Field Simulation of Temperature-Dependent Thermal Shock Fracture of Al_2_O_3_/ZrO_2_ Multilayer Ceramics with Phase Transition Residual Stress

**DOI:** 10.3390/ma16020734

**Published:** 2023-01-11

**Authors:** Yong Pang, Dingyu Li, Xin Li, Ruzhuan Wang, Xiang Ao

**Affiliations:** 1School of Civil Engineering and Architecture, Chongqing University of Science and Technology, Chongqing 401331, China; 2Chongqing Key Laboratory of Nano-Micro Composite Materials and Devices, School of Metallurgy and Materials Engineering, Chongqing University of Science and Technology, Chongqing 401331, China; 3College of Petroleum and Natural Gas Engineering, Chongqing University of Science and Technology, Chongqing 401331, China

**Keywords:** multilayer ceramics, residual stress, thermal shock, phase-field method, temperature-dependent

## Abstract

Compared with single-phase ceramics, the thermal shock crack propagation mechanism of multiphase layered ceramics is more complex. There is no experimental method and theoretical framework that can fully reveal the thermal shock damage mechanism of ceramic materials. Therefore, a multiphase phase-field fracture model including the temperature dependence of material for thermal shock-induced fracture of multilayer ceramics is established. In this study, the effects of residual stress on the crack propagation of ATZ (Al_2_O_3_-5%tZrO_2_)/AMZ (Al_2_O_3_-30%mZrO_2_) layered ceramics with different layer thickness ratios, layers, and initial temperatures under bending and thermal shock were investigated. Simulation results of the fracture phase field under four-point bending are in good agreement with the experimental results, and the crack propagation shows a step shape, which verifies the effectiveness of the proposed method. With constant thickness, high-strength compressive stress positively changes with the layer thickness ratio, which contributes to crack deflection. The cracks of the ceramic material under thermal shock have hierarchy and regularity. When the layer thickness ratio is constant, the compressive residual stress decreases with the increase in the layer number, and the degree of thermal shock crack deflection decreases.

## 1. Introduction

Ceramic materials have the advantages of high strength, high-temperature resistance, and corrosion resistance, and are widely used in aerospace, metallurgy, machinery, medical, and other fields. The development of ceramic materials in complex service environments is limited because of their inherent brittleness and poor thermal shock resistance. One of the main causes of ceramic material breakdown is thermal shock, and hence there is an urgent need to increase the ceramic materials’ resistance to thermal shock [1].

In recent years, some scholars have adopted the idea of a biomimetic composite structure design to design and prepare new layered ceramic composites for improving the thermal shock resistance of materials [2,3]. Wei et al. [4] prepared layered ZrB_2_-SiC-G ultra-high-temperature composites with weak interfacial bonding. The research results show that the thermal shock resistance of layered ceramic composites with weak interfacial bonding is greatly improved compared with bulk materials due to the remarkable increase in fracture toughness, but the fracture strength is reduced. Zhou et al. [5,6] studied the cooling thermal shock behavior and its influencing factors of layered ZrB_2_-SiC ultra-high-temperature ceramic composites with strong interface bonding through indentation quenching experiments and found that the material has higher crack growth resistance than bulk ZrB_2_-SiC ultra-high-temperature ceramic composites during thermal shock. The effect of the geometric scale on the thermal shock performance of Al_2_O_3_/ZrO_2_ multilayer ceramics was examined by Bermejo [7,8,9]. Due to the elastic mismatch between the various layers and the zirconia phase transition on the Al_2_O_3_-30%mZrO_2_ layer, the results demonstrate that the residual stress distribution curve in the laminate determines the material’s thermal shock response. Al_2_O_3_ and ZrO_2_ ceramic materials have good high radiation and heat resistance [10,11], and have good application prospects in future fusion energy. Vandeperre [12] compared and analyzed the thermal shock of composite and layered ceramic materials at different temperatures and materials through experiments, revealing the phenomenon that crack deflection prevents crack penetration. However, the above studies are based on the conclusions obtained by experimental methods, which cannot well-observe the dynamic evolution process of cracks in ceramic materials under thermal shock.

To better reveal the thermal shock damage mechanism in layered ceramic materials, researchers have developed a variety of numerical methods to simulate the initiation and propagation of thermal shock cracks, including the cohesive zone model (CZM) [13], extended finite element method (EFEM) [14], and phase-field method (PFM) [15]. The phase-field method has improved as a useful tool for examining fracture crack propagation in recent years. The crack propagation is numerically modeled by adding a globally continuous auxiliary scalar phase field that may explain both the damaged and undamaged states of the material. The smooth continuous governing equation, which may be directly solved by the finite element method or other numerical methods, converts the crack growth from a discontinuous surface problem to a volume problem. Moreover, the phase-field method can predict the initial location, propagation path, and crack branch by solving the partial differential equations, while the traditional Griffith fracture theory cannot do this [16,17]. Carollo [18] proposed a variational method suitable for the prediction of complex crack paths, including crack branching, crack deflection, and viscous stratification, and combined it with the phase-field method. For the evolution of phase-field parameters, Borden [19,20] created a higher-order equation that calls for complex finite element formulas to solve. A new model for anisotropic materials was constructed after the phase-field fracture model, which was based on the fracture behavior of isotropic materials. For brittle fracture, the elastic response and surface energy of the material are anisotropic. Hakim [21] derived a force balance condition to explain the anisotropic surface energy and found that the directed fracture energy rather than the elastic anisotropy determined the crack path. Zhang [22,23] proposed an anisotropic brittle fracture phase-field model (PFM) with a single crystal considered in terms of elastic constants and fracture energy to study the effect of anisotropic parameters on crack propagation. Nguyen [24,25] used the phase-field method for simulating the complex microstructure of interface damage and an integral formula of interactions between the brittle crack, through regularization of the interface transition zone for better describing the interface characteristics of layered ceramic. The results prove that a crack in the structure of simulated 2D and 3D interfaces is a flexible and effective interaction.

Under the finite element framework, the phase-field method has advantages in establishing the coupling fracture models of thermo-mechanical fracture, chemo-mechanical fracture, hydraulic fracture, and other multi-physical fields. Bourdin [26] studied the initiation and selective distribution of cracks in brittle materials when thermal shock was present. The outcomes of the numerical simulation closely match those of the experiment. In addition to researching the behavior of ceramic materials when subjected to thermal shock dynamic cracking, Miehe [27] suggested a completely coupled thermomechanical PFM suitable to brittle fracture. Chu [28] and Wang [29] et al. established a thermo-mechanical coupling phase-field fracture model of thermal shock for ceramic materials and simulated and analyzed the dynamic crack growth of ceramic materials under thermal shock. Although the simulation results of the phase-field model of thermal shock fracture are in good agreement with the experimental results, the numerical simulation results show that the damage evolution occurs in the non-crack region of the experimental results. Phase-field modeling ignores the material properties in the process of thermal shock with temperature changes.

In this study, a thermal-mechanical coupling PFM for multiphase material was established to investigate the thermal shock fracture of multilayer ceramic materials. To prevent unanticipated damage in the uncracked area, a temperature-dependent fracture energy threshold was included to the model. Considerations were made for temperature dependency, residual stress, and the interaction of fracture formation and heat transport. The remainder of this article is structured as follows. Section 2 introduces the PFM of thermal-mechanical coupling fracture, and the threshold of fracture energy with temperature dependence is introduced. Then, the Al_2_O_3_/ZrO_2_ multilayer ceramic model is established in Section 3. Four-point bending numerical simulation is performed on the model using the UEL and USDFLD subroutines of the finite element software ABAQUS2020, and the results are compared with the experimental findings to confirm the PFM’s validity. Section 4 evaluates the thermal shock cracking behavior of Al_2_O_3_/ZrO_2_ multilayer ceramic samples under a 2D water-quenching condition and its influencing factors are studied by using the established model and the numerical realization method. The results of the numerical simulation and the findings from the experiment show a good agreement.

## 2. Thermodynamic Coupling Damage Fracture Phase-Field Model

### 2.1. Phase-Field Approximation of Cracks

Consider an elastic body, $\Omega \subset R^{n}$ (n = 1, 2, 3, representing spatial dimensions). When n = 1, there exists an infinite 1D bar with a cross-section Γ, and the area occupied by the cross-section is Ω=Γ×L, where $L=\left[-\infty ,+\infty \right]$, and x∈L. $d\left(x\right)\in \left[0,1\right]$ is introduced as an auxiliary variable function, where *x* = 0, and *d*(*x*) = 1, indicating that the material is completely broken. When *d*(*x*) = 0, the material is intact, as the blue line in Figure 1 shows the sharp crack. In the red line part, the exponential function is used to simulate the discontinuous tip crack as a smooth continuous diffusion crack [17,30].
(1)$$d\left(x\right)=e^{-\left|x\right|/l_{c}}$$
where *l_c_* is the parameter for the regularized length scale. Dirichlet boundary conditions are satisfied by this formula: *d* (0) = 1 and *d* (±∞) = 0.

Equation (1) is the solution to the homogeneous differential Equation [17]:(2)$$d\left(x\right)-l_{c}^{2}\Delta d\left(x\right)=0$$

Equation (2) also obeys Dirichlet boundary conditions. Differential Equation (2) is the Euler equation of the variational principle:(3)$$d\left(x\right)=\mathrm{Arg}\left\{\underset{d=S}{\inf}I\left(d\left(x\right)\right)\right\}$$
where $S=\left\{d|d\left(0\right)=1,d\left(\pm \infty \right)=0\right\}$. $I\left(d\right)$ can be obtained by the integral differential Equation (2). *l*_c_ is related to the crack surface Γ, and there is a relation $I\left(d\right)=l_{c}\Gamma$ [17]. The crack surface can be written as:(4)$$\Gamma \left(d\right)=\frac{1}{l_{c}}I\left(d\right)=\frac{1}{2l_{c}}\int_{\Omega }\left(d^{2}+l_{c}{}^{2}{d^{\prime }}^{2}\right)dV$$

We introduced the unit volume crack surface energy density function, which was extended to multiple dimensions by [17]:(5)$$\Gamma \left(d\right)=\int_{\Omega }\gamma \left(d,d^{\prime }\right)dV\;\mathrm{and}\;\gamma \left(d,\nabla d\right)=\frac{1}{2l_{c}}d^{2}+\frac{l_{c}}{2}{\left|\nabla d\right|}^{2}$$

### 2.2. Fracture Variational Formula of Thermo-Mechanical Coupling Phase Field

In this section, the multiphase field coupling model related to the composition of the composite is introduced. The main purpose is to explain the fracture mode of cracks in multiphase materials by introducing multiple damage variables. Therefore, it is assumed $\Omega \subset R^{n}$ is the open domain of an isotropic thermal elastomer, where $n\in \left\{1,2,3\right\}$ represents its spatial dimension, and the outer boundary and inner discontinuous set of the elastomer can be expressed as ∂Ω and Γ, respectively, as shown in Figure 2. The crack in Figure 2a is denoted as Γ*_i_*, which can be approximated as the diffusion crack characterized by phase-field variable *d_i_* in Figure 2b. *i* is the type of material, and as two-phase materials are investigated in this study, then *i* = 1, 2. The boundaries of heat transfer and mechanical deformation problems can be denoted as $\partial \Omega =\partial \Omega_{Q}\cup \partial \Omega_{T}$ and $\partial \Omega =\partial \Omega_{t}\cup \partial \Omega_{u}$, respectively, where Q represents the heat flux. The heat flux at boundary $\partial \Omega_{Q}$ is equal to $\bar{Q}$, and *T* denotes the temperature. The traction vector applied at $\partial \Omega_{t}$ is represented by $\bar{t}$. In the yellow area, the internal heat source is designated as *q*. Since inertia has little effect on thermal shock fracture, it was not considered in this paper.

In accordance with Equation (5) in Section 2.1, the crack surface energy density function of the material can be expressed as Equations (6) and (7), where we assume $l_{c1}=l_{c2}=l_{c}$.
(6)$$\gamma_{1}\left(d_{1},\nabla d_{1}\right)=\frac{1}{2l_{c}{}_{1}}d_{1}{}^{2}+\frac{l_{c1}}{2}{\left|\nabla d_{1}\right|}^{2}$$
(7)$$\gamma_{2}\left(d_{2},\nabla d_{2}\right)=\frac{1}{2l_{c2}}d_{2}{}^{2}+\frac{l_{c2}}{2}{\left|\nabla d_{2}\right|}^{2}$$

According to Reference [27] and the principle of energy minimization, the free energy functional of thermo-mechanical coupling, *L_i_*, is:(8)$$\begin{array}{ll} L_{i}= & -\int_{\Omega }\psi_{\varepsilon }\left(\varepsilon^{e}\left(u,T\right)\right)dV-\int_{\Gamma_{c}}G_{ci}d\Gamma_{i}+\int_{\Omega }\left(\rho_{i}C_{p_{i}}\dot{T}+\nabla \cdot Q-q\right)dV\\ & +\int_{\partial \Omega_{t}}\bar{t}\cdot ud\Gamma_{i}+\int_{\Omega }b\cdot udV+\int_{\partial \Omega_{Q}}\bar{Q}\cdot nd\Gamma_{i} \end{array}$$
where ${\left\{G_{ci}\right\}}_{i=1,2}$, ${\left\{\rho_{i}\right\}}_{i=1,2}$, and ${\left\{C_{pi}\right\}}_{i=1,2}$ are the critical energy release rate, density, and specific heat of the two materials, respectively, and are the free energy of the two materials. The symbol *b* is the body force in the region. The total strain tensor is $\varepsilon =\left(\nabla u+\nabla u^{T}\right)/2$, and the thermal strain tensor is $\varepsilon_{T}=\alpha \Delta TI$, where α is the thermal expansion coefficient of materials, ΔT is the temperature difference, and *I* is the identity tensor. $\psi_{\varepsilon }\left(\varepsilon^{e}\left(u,T\right)\right)$ is the elastic energy density based on the linear strain tensor $\varepsilon^{e}\left(u,T\right)=\varepsilon -\varepsilon_{T}$.

Through the crack surface density function in Equations (6) and (7), the free discontinuity problem of sharp cracks is approximated by a continuous diffusion crack topology. The surface integral defined on the surface of a sharp crack is approximated by the volume integral, so the critical fracture energy can be approximated as:(9)$$\int_{\Gamma_{c}}G_{c}d\Gamma \approx \int_{\Omega }G_{c}\gamma \left(d,\nabla d\right)dV$$

By introducing the crack surface energy density function (Equations (6) and (7)) into Equation (9), the free energy, *L_i_*, can be modified as:(10)$$\begin{array}{ll} L_{i}=-\int_{\Omega }\psi_{\varepsilon }(\varepsilon^{c}(u,T),d_{i}) & dV-\int_{\Omega }\frac{G_{ci}}{{2l}_{c}}(d_{i}^{2}+l_{c}^{2}{|\nabla d_{i}}^{2})dV+\int_{\Omega }\left(\rho_{i}C_{p_{i}}\dot{T}+\nabla \cdot Q-q\right)dV\\ & +\int_{\partial \Omega_{q}}\bar{t}\cdot ud\Gamma_{i}+\int_{\Omega }b\cdot udV+\int_{\partial \Omega_{0}}\bar{Q}\cdot nd\Gamma_{i} \end{array}$$

The energy driving crack growth belongs to the tensile part and the compressive strain energy has little effect on crack growth. To describe the impact of the phase-field crack on material properties, the degradation function, *g*(*d*), acting on the positive part of strain energy, was developed. Therefore, the strain energy was decomposed by tension and compression:(11)$$\psi_{\varepsilon i}\left(\varepsilon^{e}\left(u,T\right),d_{i}^{}\right)=\left[g\left(d_{i}\right)+\delta \right]\psi_{\varepsilon }^{+}\left(\varepsilon^{e}\right)+\psi_{\varepsilon }^{-}\left(\varepsilon^{e}\right)$$
where $g\left(d\right)={\left(1-d\right)}^{4}$. When *d* = 1, a very small value, *δ* = 1.0 × 10^−10^, is introduced to ensure that the residual energy is kept in the complete fracture state and avoid the complete degradation of energy. ${\left\{\psi_{\varepsilon i}\right\}}_{i=1,2}$ are the strain energy densities of the two materials, respectively. The order of the monotone decline function, *g*(*d*), is 4, which makes the stiffness degrade faster due to the evolution of damage, thus revealing the brittle cracking.

To distinguish the fracture strain energy density under tension and compression, the elastic strain should be divided into tensile and compressive components [16]:(12)$$\varepsilon_{\pm }^{e}=\sum_{i}^{3}{\langle \varepsilon_{i}^{e}\rangle }_{\pm }n_{i}⊗n_{i}$$
where $\varepsilon_{+}^{e}$ and $\varepsilon_{-}^{e}$ are the tensile and compressive strain tensor, respectively. The operators ${\langle \cdot \rangle }_{\pm }=\left(\cdot \pm \left|\cdot \right|\right)/2$. The term ${\left\{\varepsilon_{i}^{e}\right\}}_{i=1,2,3}$ describes the elastic principal strain in the three principal directions. Thus, the elastic strain of tensile and compressive parts can be written as [16]:(13)$$\psi_{\varepsilon }^{\pm }\left(\varepsilon^{e}\right)=\frac{\lambda }{2}{\langle \mathrm{tr}\left(\varepsilon^{e}\right)\rangle }_{\pm }^{2}+\mu \mathrm{tr}\left(\varepsilon_{\pm }^{e}{}^{2}\right)$$
where *λ* and *µ* are Lame constants. The stress tensor introduced by definition is:(14)$$\sigma_{i}=\frac{\partial \psi_{e}}{\partial \varepsilon^{e}}=\left[g\left(d_{i}\right)+\delta \right]\left[\lambda {\langle \mathrm{tr}\left(\varepsilon^{e}\right)\rangle }_{+}I+2\mu \varepsilon_{+}^{e}\right]+\lambda {\langle \mathrm{tr}\left(\varepsilon_{}^{e}\right)\rangle }_{-}I+2\mu \varepsilon_{-}^{e}$$

In which ${\left\{\sigma_{i}\right\}}_{i=1,2}$ is the stress matrix of the two materials, respectively.

When the free energy functional in Equation (10) satisfies $\delta L_{i}=0$, a strong form of the governing equation as shown below can be given:(15a)divσ+b=0 in Ω
(15b)$$\rho_{i}C_{p_{i}}\dot{T}+\nabla \cdot Q=q\;\mathrm{in}\;\Omega$$
(15c)$$\frac{G_{ci}}{l_{c}}\left(d_{i}-l_{c}^{2}\Delta d_{i}\right)=g^{\prime }\left(d_{i}\right)\psi_{e}^{+}\;\mathrm{in}\;\Omega$$

We assume that n represents the outer normal vector pointing to the boundary, and the governing equations above obey Dirichlet and Neumann boundary conditions:(16a)$$\sigma \cdot n=\bar{t}\;\mathrm{on}\;\partial \Omega_{t}$$
(16b)$$Q\cdot n=\bar{Q}\;\mathrm{on}\;\partial \Omega_{Q}$$
(16c)$$\nabla d_{i}\cdot n=0\;\mathrm{on}\;\partial \Gamma_{c}$$

The initial conditions are as follows:(17a)$$T(x,t)=T_{0}(x,0)\;\mathrm{in}\;\Omega$$
(17b)$$u(x,t)=u_{0}(x,0)\;\mathrm{in}\;\Omega$$
(17c)$$d_{i}(x,t)=d_{i0}(x,0)\;\mathrm{in}\;\Omega$$

Here, the heat flux is controlled by temperature gradient and thermal conductivity, k, and the thermal conductivity decreases with the decrease of the phase-field variable *d*, and its degradation function, *g*(*d*), is the same as the stiffness degradation function in elastic strain energy. *k*_0_ represents the initial thermal conductivity of the material when it is undamaged.
(18)$$Q=-k\nabla T,\;k=g(d_{i})k_{0}$$

### 2.3. Thermal-Mechanical Coupling Fracture Phase Field Model with a Fracture Energy Threshold

According to Equation (15c), if the tensile strain energy, $\psi_{e}^{+}$, is non-negative, the phase-field variable *d* will change even if the value is small; that is, the stress does not need to reach the fracture strength of the material. As a result, the stiffness of the material is reduced in the undamaged area, and crack growth occurs in the non-damaged area. To avoid unexpected phase-field damage evolution, the fracture energy threshold [31] was introduced into Equation (10), and then the free energy, *L*, can be expressed as:(19)$$\begin{array}{ll} L_{i}= & -\int_{\Omega }\psi_{\varepsilon i}\left(\varepsilon^{e}\left(u,T\right),d_{i}\right)dV-\int_{\Omega }\left[1-{\left(1-d_{i}\right)}^{4}-\delta \right]\psi_{c_{i}}+\psi_{ci}\left(d_{i}{}^{2}+l_{c}^{2}{\left|\nabla d_{i}\right|}^{2}\right)dV\\ & +\int_{\Omega }\left(\rho_{i}C_{p_{i}}\dot{T}+\nabla \cdot Q-q\right)dV+\int_{\partial \Omega_{t}}\bar{t}\cdot ud\Gamma_{i}+\int_{\Omega }b\cdot udV+\int_{\partial \Omega_{Q}}\bar{Q}\cdot nd\Gamma_{i} \end{array}$$
where ${\left\{\psi_{c}^{}\right\}}_{i-1,2}$ is the threshold fracture energy for the two materials. When $\psi_{e}^{+}<\psi_{c}^{}$ is present in a material region, the phase-field variable *d* = 0 under the action of $\psi_{c}^{}$, indicating that no damage has evolved there, hence preventing the appearance of pseudo-damage evolution. Furthermore, the phase field’s governing equation, which was obtained from Equation (19), can be modified as follows to ensure the irreversibility of the crack:(20)$$\psi_{ci}\left(d_{i}-l_{c}^{2}\Delta d_{i}\right)=-2{\left(1-d_{i}\right)}^{3}H_{ei}\left(u(x,\beta ),T\right)\;\;\mathrm{in}\;\Omega$$
in which $H_{ei}\left(u(x,\beta ),T\right)=\underset{\beta \in \left[0,t\right]}{\max}\langle \psi_{e}^{+}\left(u(x,\beta ),T\right)-\psi_{ci}^{}\rangle$, ${\left\{H_{ei}\right\}}_{i=1,2}$, respectively, represents the history function of the two materials.

### 2.4. Governing Equation of Temperature-Dependent Fracture Energy Threshold

Thermal-mechanical properties of ceramics, one of the materials utilized most frequently in high-temperature applications, are directly connected to temperature. Therefore, the impact on the fracture energy threshold must be considered. According to Reference [32], the temperature-dependent fracture energy threshold has the form: (21)$$\psi_{c}\left(T\right)=\psi_{c}\left(T_{0}\right)\left[1-\frac{\int_{0}^{T}C_{p}\left(T\right)dT}{\int_{0}^{T_{m}}C_{p}\left(T\right)dT}\right]$$
where *C_p_*(*T*) is the specific heat capacity of the material at constant pressure and temperature, *T*, *T*_m_ is the melting point of the material, $\psi_{c}\left(T_{0}\right)$ can be determined by the uniaxial tensile test, and the specific expression is:(22)$$\psi_{c}\left(T_{0}\right)=\frac{{\left[\sigma \left(T_{0}\right)\right]}^{2}}{2E\left(T_{0}\right)}$$

## 3. Numerical Implementation of the PFM

### 3.1. The Discretization of Finite Elements

The governing Equations (15)–(17) are solved by the finite element method, and its weak form can be written as:(23)$$\int_{\Omega }\rho_{i}C_{p_{i}}\dot{T}\delta TdV-\int_{\Omega }Q\cdot \nabla \left(\delta T\right)dV-\int_{\Omega }q\delta TdV+\int_{\partial \Omega_{Q}}\bar{Q}\delta Td\Gamma_{i}=0$$
(24)$$-\int_{\Omega }\sigma :\delta \varepsilon dV+\int_{\partial \Omega_{t}}\bar{t}\cdot \delta ud\Gamma +\int_{\Omega }b\cdot \delta udV=0$$
(25)$$\int_{\Omega }2{\left(1-d_{1}\right)}^{3}H_{e}\delta d_{1}dV+\int_{\Omega }2\psi_{c1}\left(d\delta d_{1}^{}-l_{c}^{2}\nabla d_{1}\nabla \left(\delta d_{1}\right)\right)dV=0$$
(26)$$\int_{\Omega }2{\left(1-d_{2}\right)}^{3}H_{e}\delta d_{2}dV+\int_{\Omega }2\psi_{c2}\left(d\delta d_{2}^{}-l_{c}^{2}\nabla d_{2}\nabla \left(\delta d_{2}\right)\right)dV=0$$
where δT, δu, and $\delta d_{i}$ are the variational test functions. The finite element shape functions $N_{I}^{T}$, $N_{I}^{u}$, and $N_{I}^{d}$ for the variables *u*, *T*, and *d_i_* are introduced as:(27)$$T=\sum_{I=1}^{n}N_{I}^{T}T_{I}\;u=\sum_{I=1}^{n}N_{I}^{u}u_{I}\;d_{1}=\sum_{I=1}^{n}N_{I}^{d_{1}}d_{1}{}_{I}\;d_{2}=\sum_{I=1}^{n}N_{I}^{d_{2}}d_{2}{}_{I}$$
where *T_I_*, *u_I_*, and *d_I_* are the temperature, displacement, and phase-field values of the node *I* in an element, respectively. The symbol *n* is the number of nodes in the element. The temperature gradient, displacement gradient, and phase-field gradient can be represented, respectively, by geometric matrix *B*:(28)$$\nabla T=\sum_{I=1}^{n}B_{I}^{T}T_{I}\;\varepsilon =\sum_{I=1}^{n}B_{I}^{u}u_{I}\;\nabla d_{1}=\sum_{I=1}^{n}B_{I}^{d_{1}}d_{1I}\;\nabla d_{2}=\sum_{I=1}^{n}B_{I}^{d_{2}}d_{2I}$$

By introducing Equations (27) and (28) into Equations (23)–(26), considering the arbitrariness of the test function at the element level, the residuals of the weak forms are:(29)$$R_{I}^{T}=\int_{\Omega }\rho_{i}C_{p_{i}}\dot{T}N_{I}^{T}dV-\int_{\Omega }QB_{I}^{T}dV-\int_{\Omega }qN_{I}^{T}dV+\int_{\partial \Omega_{Q}}\bar{Q}N_{I}^{T}dA$$
(30)$$R_{I}^{u}=\int_{\Omega }\sigma B_{I}^{u}dV-\int_{\partial \Omega_{t}}\bar{t}N_{I}^{u}d\Gamma -\int_{\Omega }bN_{I}^{u}dV$$
(31)$$R_{I}^{d_{1}}=\int_{\Omega }2\psi_{c1}l_{c}^{2}B_{I}^{d_{1}}\nabla d_{1}dV+\int_{\Omega }2N_{I}^{d_{1}}\left[\psi_{c1}d_{1}-2{\left(1-d_{1}\right)}^{3}H_{e}\right]dV$$
(32)$$R_{I}^{d_{2}}=\int_{\Omega }2\psi_{c2}l_{c}^{2}B_{I}^{d_{2}}\nabla d_{2}dV+\int_{\Omega }2N_{I}^{d_{2}^{}}\left[\psi_{c2}d_{2}-2{\left(1-d_{2}\right)}^{3}H_{e}\right]dV$$

The nonlinear solver and interleaving algorithm built in ABAQUS were used to solve the multi-field coupling problem. The solution process of the corresponding Newton–Raphson method can be expressed as:(33)$${\left\{\begin{array}{c} T\\ u\\ d_{1}\\ d_{2} \end{array}\right\}}_{n+1}={\left\{\begin{array}{c} T\\ u\\ d_{1}\\ d_{2} \end{array}\right\}}_{n}-{\left[\begin{array}{cccc} K_{IJ}^{T} & 0 & 0 & 0\\ 0 & K_{IJ}^{u} & 0 & 0\\ 0 & 0 & K_{IJ}^{d_{1}} & 0\\ 0 & 0 & 0 & K_{IJ}^{d_{2}} \end{array}\right]}^{-1}{\left\{\begin{array}{c} R_{I}^{T}\\ R_{I}^{u}\\ R_{I}^{d_{1}}\\ R_{I}^{d_{2}} \end{array}\right\}}_{n}$$
where the corresponding stiffness matrix is:(34)$$K_{IJ}^{T}=\frac{\partial R_{I}^{T}}{\partial T_{J}}=\int_{\Omega }\frac{\rho_{i}C_{pi}}{\Delta t}{\left(N_{I}^{T}\right)}^{T}N_{J}^{T}dV+\int_{\Omega }{\left(1-d_{i}\right)}^{4}k{\left(B_{I}^{T}\right)}^{T}B_{J}^{T}dV$$
(35)$$K_{IJ}^{u}=\frac{\partial R_{I}^{u}}{\partial u_{J}}={\int_{\Omega }\left[{\left(1-d_{i}\right)}^{4}+\delta \right]\left(B_{I}^{u}\right)}^{T}D_{0}B_{J}^{u}dV$$
(36)$$K_{IJ}^{d_{1}}=\frac{\partial R_{I}^{d_{1}}}{\partial d_{1J}}=\int_{\Omega }2\psi_{c1}l_{c}^{2}{\left(B_{I}^{d_{1}}\right)}^{T}B_{J}^{d_{1}}dV+\int_{\Omega }2{\left(N_{I}^{d_{1}}\right)}^{T}N_{J}^{d_{1}}\left[\psi_{c1}+6{\left(1-d_{1}\right)}^{2}H_{e}\right]dV$$
(37)$$K_{IJ}^{d_{2}}=\frac{\partial R_{I}^{d_{2}}}{\partial d_{2J}}=\int_{\Omega }2\psi_{c2}l_{c}^{2}{\left(B_{I}^{d_{2}}\right)}^{T}B_{J}^{d_{2}}dV+\int_{\Omega }2{\left(N_{I}^{d_{2}}\right)}^{T}N_{J}^{d_{2}}\left[\psi_{c2}+6{\left(1-d_{2}\right)}^{2}H_{e}\right]dV$$

### 3.2. Numerical Implementation of the Finite Element Method

The above solution was implemented through the user subroutines UEL and USDFLD in ABAQUS2020. Using two element types in a hierarchical manner, a two-layer structure was employed, where layers share the same nodes but have varying degrees of freedom (DOF) [33,34]. The first type of element was a coupling of degrees of freedom for displacement and temperature. Only one DOF can be used to calculate the phase field for the second-layer element type. The field variable in the user subroutine USDFLD was set to the phase-field damage variable *d*. The user subroutine USDFLD was called on at each Gaussian point in the first layer and UEL was called on at each element in the second layer to change the properties of the material at the Gaussian integral point. Therefore, it is simple to implement the influence of the phase-field damage variable *d* on the mechanical and thermal properties.

## 4. Results and Discussions

### 4.1. PFM Simulation of the Four-Point Bending Test of Al_2_O_3_/ZrO_2_-Layered Ceramics

The crack propagation of layered ceramics under bending was studied by using the FEM. A simplified 2D PFM of crack propagation in ATZ (Al_2_O_3_-5%tZrO_2_)/AMZ (Al_2_O_3_-30%mZrO_2_)-layered ceramics under plane strain conditions was established by using commercial finite element software ABAQUS, as shown in Figure 3. The layered ceramic specimen was composed of five layers of ATZ and four layers of AMZ, alternately. The geometric size was 60 × 3 mm and the thickness of the three models remained constant. The layer thickness ratio, R (ATZ layer thickness/AMZ layer thickness), was 2, 5, and 10. The effect of residual stress on the crack propagation of layered ceramics with different layer thickness ratios was considered. The material parameters for each layer are shown in Table 1 and Table 2. This model does not consider the change of porosity and its influence on the material during the loading process. For further research, please refer to Reference [35].

According to References [8,36], layered ceramics are processed by special technology, and residual stress is observed after sintering and cooling, which has a certain influence on the crack growth of layered ceramics under bending. Therefore, we applied a prestress in the x-direction in the finite element model of layered ceramics. The magnitude of the prestress was obtained by using the FEM proposed in this study. As shown in Figure 4, it was basically consistent with the theoretical value, which verifies the effectiveness of the method. As shown in Figure 5a, the beginning and spreading of cracks in layered ceramics under a four-point bending load were realized by using the FEM derived in this study. The crack deflection occurred when the crack propagated to the weak interface layer, and the propagation path presented a stepped shape, which was well in line with the experimental results in the literature (see Figure 5b). No accidental damage in the simulation results inconsistent with the experimental results was observed, which further proves the rationality of the temperature-dependent fracture energy threshold model.

Under the same thickness of the specimen, the thickness of the AMZ layer decreased with the increase in the layer thickness ratio, and the ATZ showed the opposite. The size of the specimen had a great influence on the stress distribution of each layer of the layered ceramic. As shown in Figure 4, the residual stress of the tensile layer gradually decreased with the increase in the layer thickness ratio, ranging from 58 to 234 MPa, whereas the residual stress of the compressible layer gradually increased, ranging from −722 to −604 MPa. Figure 6 shows the stress distribution in each layer of the finite element model of Al_2_O_3_/ZrO_2_-layered ceramics with different layer thickness ratios, which was consistent with that described in Reference [37]. In terms of residual stress, the layered ceramic with R = 10 was the optimal design, which combines the minimum tensile stress of the surface layer and the maximum compressive stress of the inner layer.

Numerical simulations were conducted by using the above method with the model established to investigate the effect of residual stress and the layer thickness ratio on the crack propagation of Al_2_O_3_/ZrO_2_-laminated ceramic specimens under bending. The results are shown in Figure 7. Under the same layer thickness ratio, the model crack with residual stress greatly deflected when it extended to the weak interface layer, whereas the model crack without residual stress deflected less when it extended to the weak interface layer, or did not deflect (such as R = 2). With the increase in the layer thickness ratio, cracks were deflected to a greater extent at the weak interface, where the degree of deflection refers to the distance between longitudinal extension cracks in two adjacent ATZ layers, with the weak interface layer being the AMZ layer. The high compressive stresses in the weak interface layer allowed the cracks to deflect between the layers, effectively avoiding catastrophic damage to the specimen under the action of external forces.

The force–displacement relationship for Al_2_O_3_/ZrO_2_-laminated ceramic specimens with residual stresses under bending is shown in Figure 8. When the layer thickness ratio R = 2, the crack started to deflect with a small displacement, the smaller compressive stress in the compressed layer made the crack deflect less, and the crack unloaded with a small force when it penetrated the weak interfacial layer, so the ‘stepped’ unloading pattern was observed, as shown in the force–displacement diagram. When the layer thickness ratio increased, the maximum load that the layered ceramic material can withstand increased, and the deflection degree of the crack propagation to the weak interface layer increased under the action of high-strength compressive stress. Therefore, the crack deflection occurred during the unloading process. When the displacement was smaller, the unloading speed was faster and the unloading gradient was larger, such as R = 10.

### 4.2. PFM Simulation of the Thermal Shock Fracture of Al_2_O_3_/ZrO_2_-Laminated Ceramic Specimens under Water Quenching

A 2D phase-field fracture model was established by using the proposed FEM to study the effects of the layer thickness ratio and residual stress on the thermal shock crack propagation of Al_2_O_3_/ZrO_2_-laminated ceramic specimens, as shown in Figure 9. Considering the symmetry of the model, taking one-half of the specimen for modeling and using symmetric boundary conditions are necessary. The geometric size of the model was 25 × 3 mm (the red box shows the actual modeling). The temperature dependence of the thermal conductivity, thermal expansion coefficient, specific heat capacity, and elastic modulus was considered in the material parameters, as shown in Figure 10. The thermal expansion coefficient of the AMZ layer changed abruptly at 700 °C, which is the result of the phase transformation of ZrO_2_, whereas the thermal expansion coefficient of the ATZ layer changed minimally between 20 and 1200 °C. To facilitate the convergence of the calculation results, we set the elastic modulus of elasticity of the material after damage to 0.0001 *E*_0_ (*E*_0_ is the initial modulus of elasticity of the material without damage), the conductivity to 0.1 *K*_0_ (*K*_0_ is the conductivity before damage), the water-quenching temperature of the thermal shock to 20 °C, and the heat exchange coefficient to h_f_ = h_v_ = 70 kW/(m^2^·K).

In Figure 11, the left-hand side shows the specimen with residual stresses after the sintering and cooling process, and the right-hand side shows the ideal state of the specimen without residual stresses after processing. At an initial temperature of 300 °C, the number of thermal shock cracks in the surface layer increased with the increasing layer thickness ratio, regardless of the presence of residual stresses inside the specimen. As mentioned previously, with the increase of the layer thickness ratio, the residual stress of the tensile layer gradually decreased, whereas that of the compressible layer gradually increased with the increase in the layer thickness ratio. Under the action of thermal shock, the surface crack of the specimen passed through the first tensile layer to the second compressive layer. Under the high compressive stress of the compressive layer, the stress at the crack tip gradually decreased, so it could not penetrate the interface between the tensile and compressive layers of the next layer. The crack deflection began to propagate laterally in the compressive layer, such as when R = 5. The specimen with R = 10 had a relatively short length of crack transverse propagation compared with the other two thickness ratios, which was the same with or without residual stress.

Interestingly, the number of surface cracks was higher in specimens without residual stress compared to those with residual stress under the same layer thickness ratio, suggesting that the residual stress in the compressed layer somehow inhibited the production of surface microcracks. The transverse crack propagation length of the specimen without residual stress was shorter than that with residual stress, but the deflection degree of the extended crack was smaller without the action of high-strength compressive stress, and even the deflection did not occur locally, resulting in brittle failure of the specimen, such as R = 10. This finding shows that the existence of high-strength compressive residual stress can help with crack deflection and reduce the occurrence of catastrophic damage. The existence of residual stress and the layer thickness ratio changed the structural form of the interface between strong and weak materials, and the deflection phenomenon mainly occurred at the junction of the interface between strong and weak, indicating that the interface structure formed by the material was the main factor affecting the crack growth path. In terms of thermal shock, it is obvious that the damage degree of the specimen with R = 10 was smaller than that of the other two specimens in the same thermal shock environment, so the specimen with R = 10 was the optimal design.

Thermal shock is an important cause of the brittle failure of high-temperature ceramic materials. As one of the promising materials in the field of high-temperature materials, ceramic materials have complex and harsh service environments, and bionic-layered ceramic materials with excellent performance compared to single-phase ceramic materials have been widely used, gradually. The thermal shock mechanism of multiphase-layered ceramic materials is more complex, and there are few studies conducted in this field. Therefore, the thermal shock damage behavior of layered ceramic materials must be investigated to reveal the thermal shock damage mechanism of layered ceramic materials.

As shown in Figure 12, the number of thermal shock cracks in the laminated ceramic material increased with the initial temperature under a constant layer thickness ratio, and the thermal shock cracks in the surface layer showed a certain regularity and hierarchy, which was consistent with the previously studied thermal shock cracking pattern in single-phase ceramic materials [32,38,39]. When the temperature reached 300 °C, the surface crack passing through the first tensile layer was 1, and the crack deflected in the first compressive layer and began to propagate laterally to the end of the specimen. When the initial temperature rose to 600 °C, the number of crack-penetrating layers increased, but the two penetrating cracks were not in the same vertical direction, which avoided catastrophic brittle failure of layered ceramic materials in the high-temperature environment and proved the application potential of layered ceramic materials in the high-temperature field.

The influence of different layers on the thermal shock cracks of laminated ceramics with a fixed layer thickness ratio was studied. As shown in Figure 13, the thermal shock crack decreased with the increase in layer numbers when the initial temperature was 300 °C. When the initial temperature reached 500 °C, the thermal shock surface crack did not change remarkably with the increase in layer numbers, but the thermal shock crack propagation resistance increased with the increase in layer numbers. These phenomena are related to the residual stress, as shown in Figure 14. The tensile residual stress in laminated ceramics increased with the increase in the number of layers, whereas the compressive residual stress showed the opposite. With the increase in the number of layers, the compressive residual stress at the weak interface gradually decreased, and the stress-weakening ability at the crack tip decreased. The weak interface layers were continually penetrated by the thermal shock fracture, and layered ceramics became less resistant to thermal shock as more layers were added.

## 5. Conclusions

The thermal shock fracture of multilayer ceramic materials was modeled using a multiphase thermo-mechanical coupling phase-field method. A temperature-dependent fracture energy threshold was included to the model to eliminate accidental damage evolution. The phase-field method and finite element method were used for numerical simulation. Firstly, the feasibility of the proposed PFM was verified, and the crack growth behavior of Al_2_O_3_/ZrO_2_-layered ceramic materials with different layer thickness ratios and without residual stress was studied under the four-point bending test. The results showed that the compressive residual stress of the AMZ layer increased with the increasing layer thickness ratio, whereas the tensile residual stress showed the opposite. The existence of the residual stress is conducive to the deflection of the cracks in the layered ceramic material. The simulated crack morphology is in good agreement with the experimental results. The results of the PFM simulation can accurately depict how cracks begin to form and propagate throughout multilayer ceramic materials during four-point bending tests. The thermal shock behavior of Al_2_O_3_/ZrO_2_-layered ceramic samples was numerically simulated. It was found that the effect of residual stress on the thermal shock crack of laminated ceramic material is consistent with that of the four-point bending test. The high compressive residual stress of AMZ can effectively avoid catastrophic failure of the material. The number of thermal shock cracks increased with the layer thickness ratio. In addition, the thermal shock damage fracture behavior of Al_2_O_3_/ZrO_2_-layered ceramic materials with the same thickness ratio and different numbers of layers was investigated at different initial temperatures. The number of thermal shock cracks and the number of layers penetrated by the cracks increased with the increase in the initial temperature, but the positions of adjacent penetrated cracks were not in the same vertical direction, effectively avoiding brittle fracture of the layered ceramic materials. For the same layer thickness ratio, the compressive residual stress decreased with the increasing number of layers and the thermal shock resistance of laminated ceramics decreased with the increasing number of layers. This study makes a good contribution to further revealing the thermal shock damage mechanism and crack propagation behavior of laminated ceramic materials.

## Figures and Tables

**Figure 1 materials-16-00734-f001:**
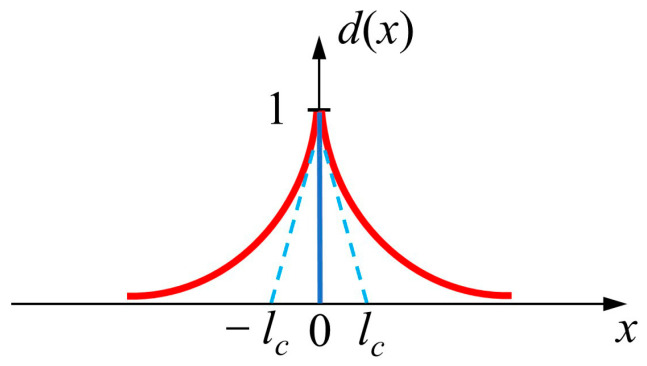
Modeling the sharp crack at *x* = 0 and the diffusion crack represented by Equation (1).

**Figure 2 materials-16-00734-f002:**
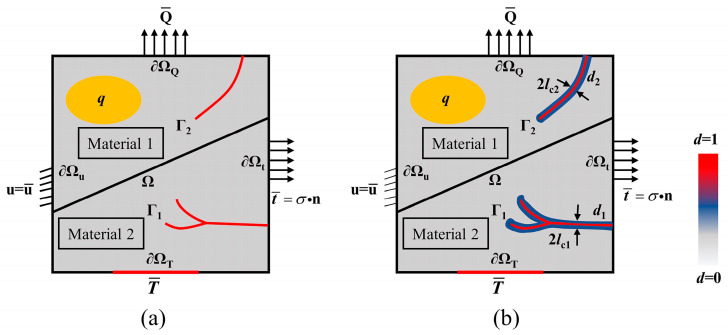
Diagram of the multiphase thermo-mechanical coupling phase-field fracture of an elastic solid. (**a**) Solid embedded sharp crack/interface geometry, (**b**) Phase field regularized crack/interface geometry.

**Figure 3 materials-16-00734-f003:**
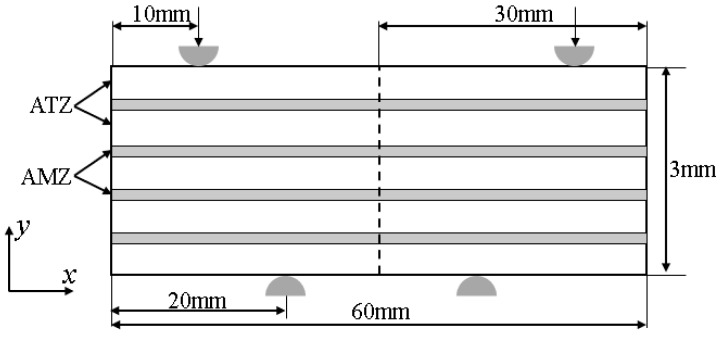
Geometric model diagram of four-point bending test for layered ceramics.

**Figure 4 materials-16-00734-f004:**
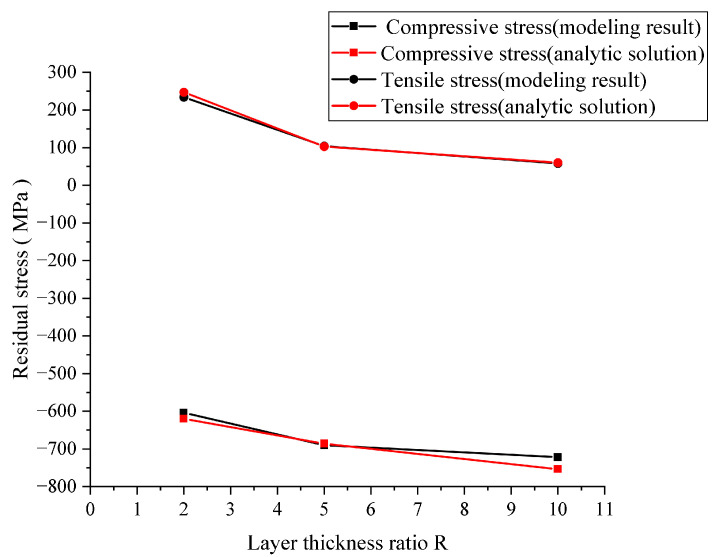
Theoretical values and numerical simulation results of residual stress in layered ceramics.

**Figure 5 materials-16-00734-f005:**
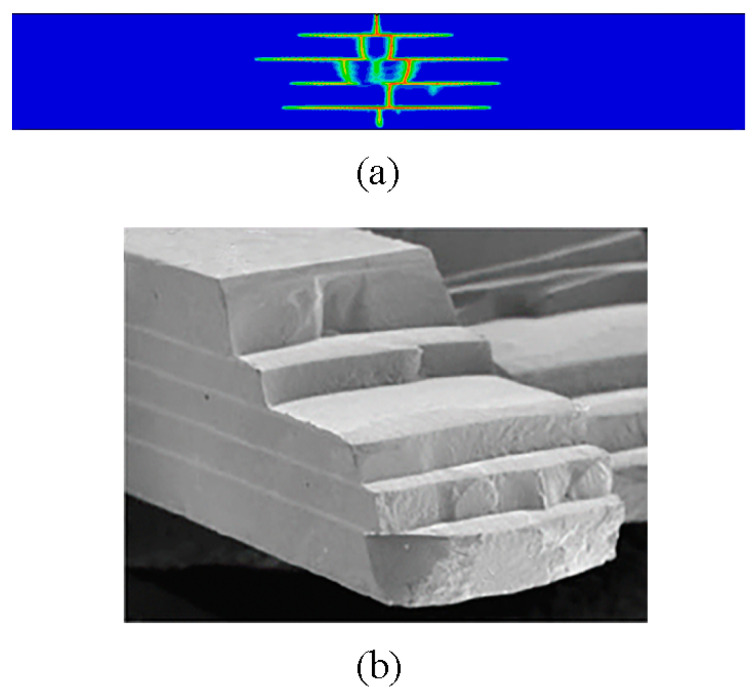
Comparison between the numerical simulation results (**a**) of four-point bending and experimental results (**b**) when the layer thickness ratio was R = 5 [37].

**Figure 6 materials-16-00734-f006:**
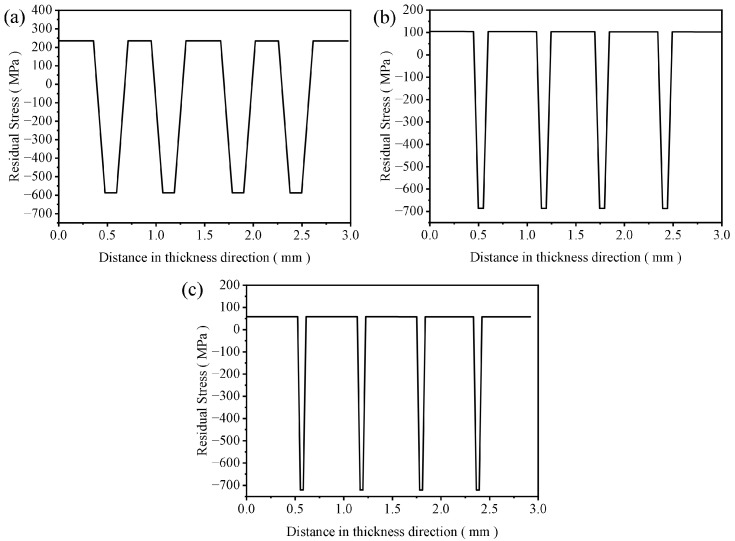
Stress distribution in the Y-direction of different layer thickness ratio models ((**a**) R = 2, (**b**) R = 5, and (**c**) R = 10).

**Figure 7 materials-16-00734-f007:**
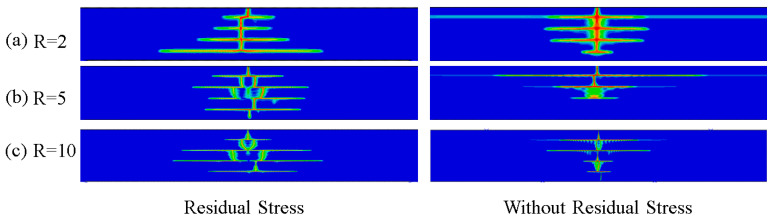
Numerical simulation results of the four-point bending of laminated ceramics with different layer thickness ratios.

**Figure 8 materials-16-00734-f008:**
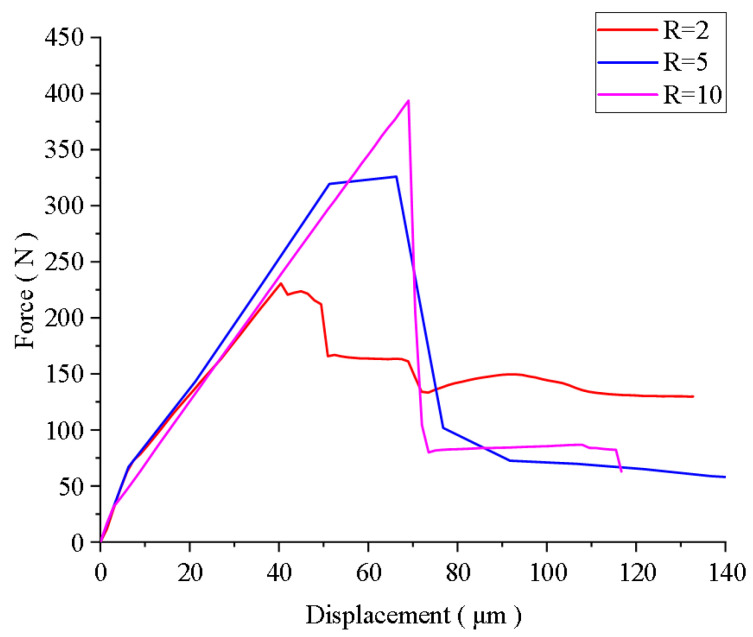
Force–displacement curve of the Al_2_O_3_/ZrO_2_-laminated ceramic specimen under bending.

**Figure 9 materials-16-00734-f009:**
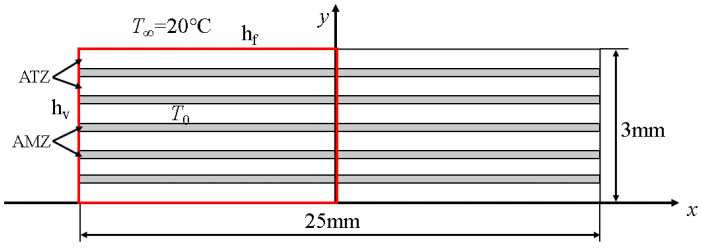
Thermal shock geometric model sketch of the Al_2_O_3_/ZrO_2_-laminated ceramic specimen.

**Figure 10 materials-16-00734-f010:**
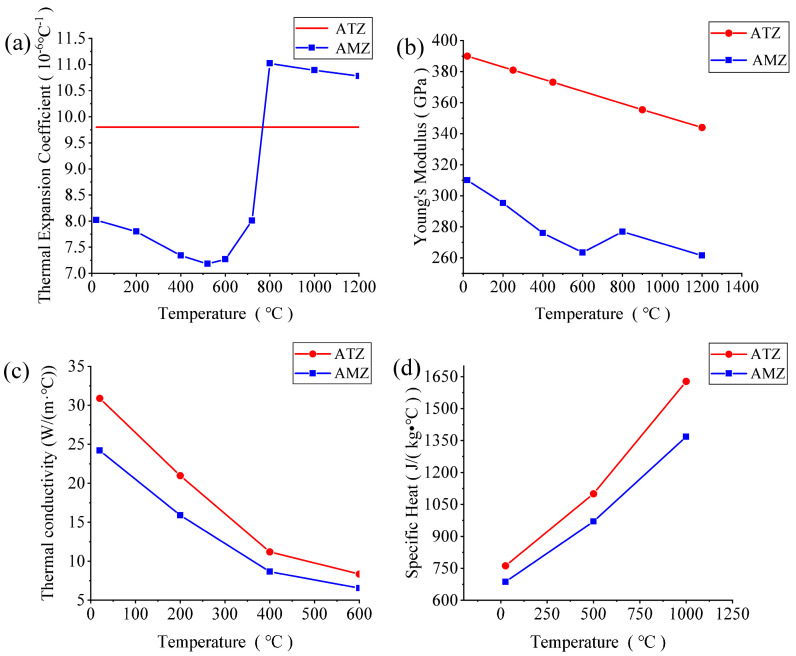
Curves of the (**a**) thermal expansion coefficient, *α*, (**b**) Young’s modulus, *E*, (**c**) thermal conductivity, *k*, and (**d**) specific heat, C*_p_*, of Al_2_O_3_/ZrO_2_-laminated ceramics versus temperature.

**Figure 11 materials-16-00734-f011:**
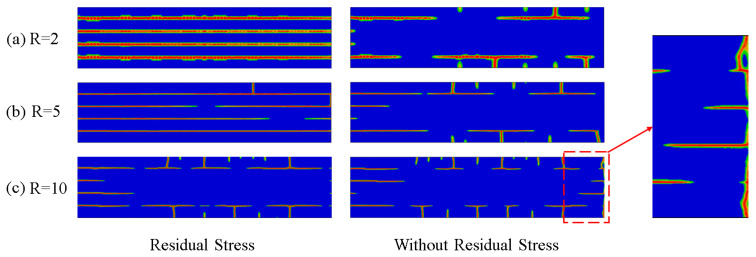
Numerical simulation results of thermal shock with different layer thickness ratios at an initial temperature of 300 °C.

**Figure 12 materials-16-00734-f012:**
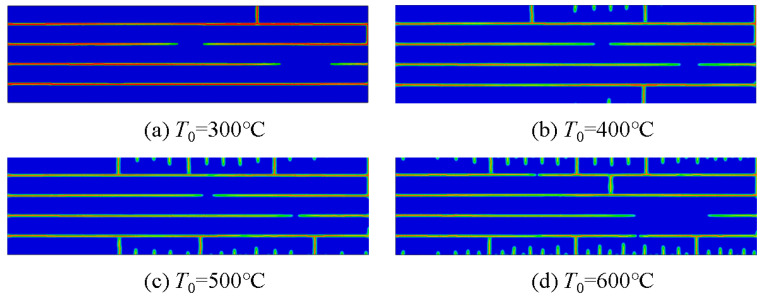
Numerical simulation results of thermal shock for layered ceramics with layer thickness ratio R = 5 at different initial temperatures.

**Figure 13 materials-16-00734-f013:**
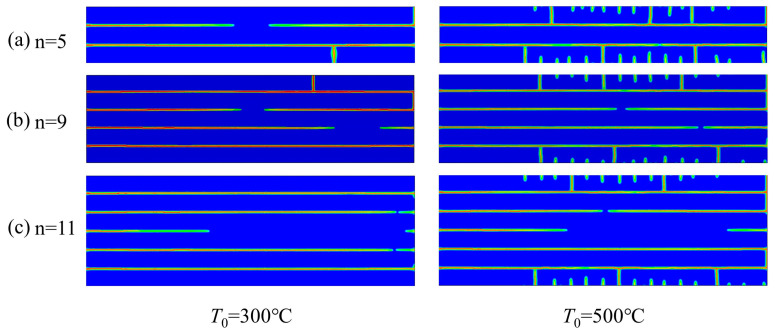
Numerical simulation results of thermal shock at different initial temperatures for layered ceramics with different layer thickness ratios, R = 5.

**Figure 14 materials-16-00734-f014:**
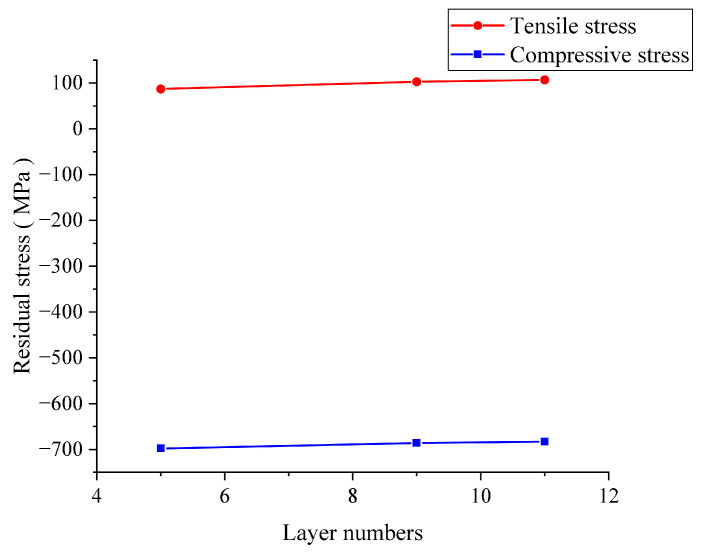
Residual stress of layered ceramics with different layer numbers.

**Table 1 materials-16-00734-t001:** Material parameters of Al_2_O_3_/ZrO_2_-layered ceramics [8].

Property	AMZ	ATZ
*E* (GPa)	310	390
*ν*	0.22	0.22
*ρ* (g/cm^3^)	4.48	4.03
*α* (10^−6^ °C^−1^)	8.02	9.8
*k* (W/(m·°C))	24.2	30.9
*C*_p_ (J/(kg·°C))	687	762
*σ* (MPa)	110	392

**Table 2 materials-16-00734-t002:** Thickness parameters of Al_2_O_3_/ZrO_2_-layered ceramics with different layer thickness ratios (layer thickness unit: mm).

Layer Thickness Ratio, R (ATZ/AMZ)	2	5	10
ATZ	0.429	0.530	0.556
AMZ	0.214	0.100	0.056

## Data Availability

The data presented in this study are available on request from the corresponding author data.

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
