# Peer review of "Phase-Field Simulation of Temperature-Dependent Thermal Shock Fracture of Al2O3/ZrO2 Multilayer Ceramics with Phase Transition Residual Stress"

_materials, 2023, doi:10.3390/ma16020734_

Round 1

Reviewer 1 Report

Title: Phase field simulation of temperature-dependent thermal shock facture of Al2O3/ZrO2 multilayer ceramics with phase transition residual stress

Authors: Yong Pang, Dingyu Li, Xin Li, Ruzhuan Wang, and Xiang Ao

The manuscript entitled “Phase-field simulation of temperature-dependent thermal shock facture of Al2O3/ZrO2 multilayer ceramics with phase transition residual stress” by Yong Pang et al. presents the phase field model simulation tool to determine the temperature-dependent thermal shock of Al2O3/ZrO2 multilayer ceramics. The effects of residual stress on the crack propagation of ATZ/AMZ multilayer ceramics with different thickness ratios, layers, and initial temperatures under bending and thermal shock were investigated. The manuscript could be recommended for publication with some minor modifications.  

Q1: In table 2, what is the unit of the layer thickness?

Author Response

Q1: In table 2, what is the unit of the layer thickness?

Response: We feel great thanks for your professional review work on our article. According to your nice suggestions, we have added the description in the table, and the detailed modification is shown in the manuscript.

Reviewer 2 Report

Reviewer Comment
Manuscript number: Materials- 212557

Dear Editor,The manuscript is successful to implement phase field method (PFM) and finite element method (FEM) in studying the crack growth behaviour of
Al2O3/ZrO2 layered ceramic materials with different layer thickness ratio and without residual stress under the four-point bending test.

The following suggestions could further improve the quality of the submitted manuscript:

1-Line 63: glob-ally should corrected to globally.

2-Line 74: mater-ials, same as previous.

3-the quality of figure 2 needed to be clearer.

4-figure 5 is repeated twice. the second figure 5 should be figure 6.

5-The authors claim they make a comparison between simulation, numerical with that experimental values (trend) in the literature. However, no any rigorous examples and references were given to support this claim.

Author Response

Q1-Line 63: glob-ally should corrected to globally.

Response: We feel sorry for our carelessness. In our resubmitted manuscript, the typo is revised. Thanks for your correction.

Q2-Line 74: mater-ials, same as previous.

Response: We were really sorry for our careless mistakes. Thank you for your reminder.

Q3-the quality of figure 2 needed to be clearer.

Response: We agree with the reviewer’s assessment. Accordingly, we have changed the picture of the manuscript.

Q4-figure 5 is repeated twice. the second figure 5 should be figure 6.

Response: We have carefully checked the manuscript and corrected the errors accordingly.

Q5-The authors claim they make a comparison between simulation, numerical with that experimental values (trend) in the literature. However, no any rigorous examples and references were given to support this claim.

Response: Thank you very much for your suggestions. We compare the experimental results with the numerical simulation results in Line 329 and 348 and Figure 5(b), and verify the accuracy of the model by referring to reference [37].

Reviewer 3 Report

Referee Report on “Phase field simulation of temperature-dependent thermal shock fracture of Al2O3/ZrO2 multilayer ceramics with phase transition residual stress

This is, of course, a work that could be recommended for publication, but only after some of the improvements formulated below.

1.       Both materials and their combinations are highly radiation and thermally resistant and are on the priority list of materials for future fusion energy applications. See, for example:

Ananchenko, D. V., Nikiforov, S. V., Sobyanin, et al (2022). Paramagnetic Defects and Thermoluminescence in Irradiated Nanostructured Monoclinic Zirconium Dioxide. Materials15(23), 8624.

Antuzevics, A., Elsts, E., Kemere, M., et al (2023). Thermal annealing of neutron irradiation generated paramagnetic defects in transparent Al2O3 ceramics. Optical Materials135, 113250.

It would be useful to reflect this fact in the introduction

2.       Lines 31-36. This paragraph needs supporting references.

3.       Something must be said about the differences in the formation of defects (defects) and how this occurs not only in the ceramics themselves, but also on the interface.

4.       Chapter 2 could be shorter. It is recommended to put excess mathematics to the supplement.

5.       What fundamentally new data on the materials under consideration were obtained in this work?

6.       Are there requirements for the purity of ceramics?

7.       How much is their porosity acceptable?  How can porosity change in the process? See, for possible processes in recent MDPI paper:

Klym, H., et al (2021). Positron annihilation lifetime spectroscopy insight on free volume conversion of nanostructured MgAl2O4 ceramics. Nanomaterials11(12), 3373.

Author Response

Q1. Both materials and their combinations are highly radiation and thermally resistant and are on the priority list of materials for future fusion energy applications. See, for example:

Ananchenko, D. V., Nikiforov, S. V., Sobyanin, et al (2022). Paramagnetic Defects and Thermoluminescence in Irradiated Nanostructured Monoclinic Zirconium Dioxide. Materials, 15(23), 8624.

Antuzevics, A., Elsts, E., Kemere, M., et al (2023). Thermal annealing of neutron irradiation generated paramagnetic defects in transparent Al2O3 ceramics. Optical Materials, 135, 113250.

It would be useful to reflect this fact in the introduction

Response: As suggested by the reviewer, we have added more references to support this idea (such as literature [10,11]). For details, see line 52-53.

Q2. Lines 31-36. This paragraph needs supporting references.

Response: We sincerely appreciate the valuable comments. We have added references to the paper. See line 36 for detailed changes.

Q3. Something must be said about the differences in the formation of defects (defects) and how this occurs not only in the ceramics themselves, but also on the interface.

Response: Thank you for your valuable suggestions. To solve these problems, we have made detailed modifications in lines 428-432 of the paper.

Q4. Chapter 2 could be shorter. It is recommended to put excess mathematics to the supplement.

Response: We are grateful for your suggestions, which have been properly condensed in the article, keeping the necessary parts.

Q5. What fundamentally new data on the materials under consideration were obtained in this work?

Response: In this paper, a theoretical framework of multiphase ceramic phase field fracture is proposed. Through this theoretical framework, influence factors (layer number, layer thickness ratio, residual stress) and thermal shock damage of Al2O3/ZrO2 layered ceramic materials are studied, and the temperature correlation between the two materials is considered. Some suggestions are provided for improving the thermal shock resistance of ceramic materials and optimizing the design of layered ceramic structures. Thank you very much.

Q6. Are there requirements for the purity of ceramics?

Response: In response to your question, we have introduced the reference source of data in Table 1. The experiment conducted in the literature considers the purity of materials, and the numerical simulation is carried out through the experimental data, so the purity of materials is considered in our model.

Q7. How much is their porosity acceptable?  How can porosity change in the process? See, for possible processes in recent MDPI paper:

Klym, H., et al (2021). Positron annihilation lifetime spectroscopy insight on free volume conversion of nanostructured MgAl2O4 ceramics. Nanomaterials, 11(12), 3373

Response: The material data in our paper comes from the experimental results in the literature. The experimental model considers the reasonable porosity of the material, but we do not discuss the influence of other porosity and porosity change in the thermal shock process. Moreover, the model considering porosity change will be more complex, and we hope to make further improvement in the future. Thank you very much for your suggestions and questions. We have made some modifications to the paper and quoted relevant references [35]. For details, please refer to lines 312-313.

Round 2

Reviewer 3 Report

After successful revision, this manuscript can be recommended for publication.

There is only need to check first surname in reference [12] 

Author Response

Comments 1: There is only need to check first surname in reference [12].

Response: We feel sorry for our carelessness. We have carried out corresponding checks and modifications in the paper.